Digital Accessible Knowledge and well-inventoried sites for birds in Mexico: baseline sites for measuring faunistic change

Peterson A. Townsend town@ku.edu 1
Navarro-Sigüenza Adolfo G. 2
Martínez-Meyer Enrique 3
1 Biodiversity Institute, University of Kansas , Lawrence , KS , United States
2 Museo de Zoología, Facultad de Ciencias, Universidad Nacional Autónoma de México , México, Distrito Federal , México
3 Instituto de Biología, Universidad Nacional Autónoma de México , México, Distrito Federal , México
Costello Mark
Electronic publication date: 2016 Sep 7
Publication date: 2016
Volume: 4
Electronic Location ID: e2362
Received 2016 Jun 14; Accepted 2016 Jul 23
Copyright: ©2016 Peterson et al.
Copyright year: 2016
Copyright holder: Peterson et al.
License: This is an open access article distributed under the terms of the Creative Commons Attribution License, which permits unrestricted use, distribution, reproduction and adaptation in any medium and for any purpose provided that it is properly attributed. For attribution, the original author(s), title, publication source (PeerJ) and either DOI or URL of the article must be cited.
License URL: https://creativecommons.org/licenses/by/4.0/

Keywords: Biodiversity, Biodiversity change, Faunal dynamics, Historical surveys, Resurveys

Funding: CONACyT 152060 Fulbright Specialists Coordinación de Investigación, Instituto de Biología, and Facultad de Ciencias, all of the Universidad Nacional Autónoma de México, and a CONACyT grant (152060) supported ATP’s research visit to Mexico, during which time this analysis was developed. Final stages of development of this analysis and manuscript were supported by a Fulbright Specialists Grant to ATP, during which time he was hosted by the Wildlife Institute of India. The funders had no role in study design, data collection and analysis, decision to publish, or preparation of the manuscript.

==============================
Background

Faunal change is a basic and fundamental element in ecology, biogeography, and conservation biology, yet vanishingly few detailed studies have documented such changes rigorously over decadal time scales. This study responds to that gap in knowledge, providing a detailed analysis of Digital Accessible Knowledge of the birds of Mexico, designed to marshal DAK to identify sites that were sampled and inventoried rigorously prior to the beginning of major global climate change (1980).

Methods

We accumulated DAK records for Mexican birds from all relevant online biodiversity data portals. After extensive cleaning steps, we calculated completeness indices for each 0.05° pixel across the country; we also detected ‘hotspots’ of sampling, and calculated completeness indices for these broader areas as well. Sites were designated as well-sampled if they had completeness indices above 80% and >200 associated DAK records.

Results

We identified 100 individual pixels and 20 broader ‘hotspots’ of sampling that were demonstrably well-inventoried prior to 1980. These sites are catalogued and documented to promote and enable resurvey efforts that can document events of avifaunal change (and non-change) across the country on decadal time scales.

Conclusions

Development of repeated surveys for many sites across Mexico, and particularly for sites for which historical surveys document their avifaunas prior to major climate change processes, would pay rich rewards in information about distributional dynamics of Mexican birds.

Introduction

The temporal dynamics of geographic distributions of species and composition of local biotic communities are central to much of biogeography, macroecology, and conservation biology (Lavergne et al., 2010). That is, how species’ distributions evolve through time and how community composition changes as a result are key in determining essentially all results in these areas of ecology and evolutionary biology. Although biogeographers invest fundamentally in retracing the geography of evolving lineages over long periods of time, strangely little information exists on short-term dynamics of species’ distributions and community composition (e.g., Nunes et al., 2007; Tingley & Beissinger, 2009).

An important opportunity to understand these shorter-term dynamics of distributions and communities by means of longitudinal comparisons of inventories of local faunas and floras. That is, when a baseline of solid, complete, and well-documented knowledge exists about a site (Colwell & Coddington, 1994; Peterson & Slade, 1998; Soberón & Llorente, 1993), re-surveys over years, decades, and centuries can offer a fascinating view into the natural dynamics of species’ ranges and the effects of human presence and activities. Drivers of these changes may act both on local scales (e.g., effects of land use change) and on global scales (e.g., effects of climate change), and potentially may interact as well; studies integrating and comparing effects of different such drivers are particularly rare (e.g., Peterson et al., 2015; Rubidge et al., 2011). Of course, detailed documentation of species identifications and of the completeness of inventory efforts are necessary for both the baseline and the resurvey, but the general paradigm has considerable potential.

Previous baseline/re-survey efforts have yielded fascinating information about faunas and floras. For example, Nilsson, Franzén & Jönsson (2008) documented 40% extirpation of butterfly species over a 90+ year span on a plot in southern Sweden, and found that species disappearing from the fauna tended to be those with a short flight length period, narrow habitat breadth, and small distributional area in Europe; however, only flight length period was significant in multivariate analyses. Grixti & Packer (2006) studied bee communities at a site in southern Ontario, and documented community changes and diversity increases over a 40+ year span, likely in response to successional changes in the surrounding landscapes. Scholes & Biggs (2005) proposed a biodiversity intactness index, and showed widespread population declines, particularly among mammals, and ecosystem declines concentrated in grasslands, across a large region of southern Africa. In California, important efforts have been carried out to resurvey sites studied by Joseph Grinnell a century ago, detecting fascinating distributional (Tingley & Beissinger, 2009), phenotypic (Leache, Helmer & Moritz, 2010), and genetic (Rubidge et al., 2011) changes in vertebrate species and communities (Moritz et al., 2008).

Within Mexico, such before-and-after studies have been particularly scarce. Peterson & Navarro-Sigüenza (2005) used nineteenth-century sources to reflect on changes in the avifauna of the Valley of Mexico over the twentieth century. In a particularly interesting example, Olvera-Vital (2012) re-surveyed the avifauna of Misantla, Veracruz—of great interest is that the avifauna has been quite stable over the past 50–100 years, which apparently reflects the very early mass-disturbance to the natural habitats of that region (Sánchez, 1998), such that the baseline inventory was itself already post-disturbance. On broader and coarser spatial scales, Peterson et al. (2015) assessed countrywide changes in Mexican endemic bird species’ distributions between the middle of twentieth century and early twenty-first century, and found dominant effects of changes in temperature (and not of human impact on landscapes or changes in precipitation) in driving avifaunal change. Curiously, however, to our knowledge at least, this short list includes all sites that have seen baseline and re-survey inventories of birds in the country, such that the nature and pattern of avifaunal change across Mexico remains very poorly characterized.

The purpose of this paper is to stimulate and enable a next generation of such re-survey efforts for birds across Mexico by means of cataloguing sites for which solid documentation exists for the original baseline inventory, and for which the baseline inventory is demonstrably complete. We reviewed all existing Digital Accessible Knowledge (DAK; Sousa-Baena, Garcia & Peterson, 2013) regarding the birds of Mexico: we probed four major online data portals for relevant data (i.e., bird records from Mexico prior to 1980), and evaluated inventory completeness at two spatial extents (0.05° grid squares, and coarser hotspots of sampling). We present a simple list and catalog of sites that have seen relatively complete (i.e., ≥80% of avifauna known and documented) inventories, as a challenge and stimulus to the ornithological community in Mexico. Resurvey of these localities would provide rich and informative rewards in understanding the dynamics of bird populations and distributions across the country.

Materials & Methods

This analysis is based on a suite of assumptions about data quality and appropriate resolutions inherent in biodiversity data. For instance, we used single days as the unit of sampling effort throughout our analyses, as this resolution has proven an effective balance between too much and too little resolution; temporal information finer than the level of day is rarely available with older biodiversity data, whereas coarse temporal resolution can underappreciate the efficacy of short-term, intensive inventory efforts. Hence, a basic working data set was the combination of species identification, day (i.e., unique combinations of year, month, and day), and place. This latter we defined in various ways, described below, but we note that we used both geographic coordinates and textual locality descriptions to avoid loss of information owing to lack of georeferencing, at least to every extent possible.

Data sources and quantities

We downloaded data from four basic sources for this study: the Global Biodiversity Information Facility (GBIF; http://www.gbif.org), VertNet (http://www.vertnet.org/), the Red Mundial de Información de la Biodiversidad (REMIB; http://www.conabio.gob.mx/remib/doctos/remib_esp.html), and UNIBIO (http://unibio.unam.mx/). Each of these biodiversity information networks has its respective strengths and weaknesses, and considerable overlaps exist among them in coverage of biodiversity information sources. We downloaded data on bird occurrences in Mexico from all four, and trusted that duplication would be removed in our data cleaning steps; for this reason, we are unable to develop comparisons among the different data resources. For GBIF and VertNet, automated download was possible; however, for REMIB and UNIBIO, we requested and were provided with data ‘dumps,’ as some of the data were restricted from full public access (REMIB) or not yet fully available to the public (UNIBIO). The full set of sources contributing data to this analysis is provided in the Appendix.

Figure 1 Digital accessible knowledge of bird distributions (481,409 unique combinations of species × place × time) across Mexico prior to 1980, drawn from GBIF, VertNet, REMIB, and UNIBIO (records are not coded by source owing to frequent overlaps among sources in serving copies of the same record).

Table 1 Summary of initial data downloaded from each of four biodiversity data portals for Mexican vertebrate classes, and the relative redundancy of records in each, at the level of species × time (year, month, day) × place (geographic coordinates, textual descriptions).

Note that subsequent data cleaning steps changed these initial tallies of redundancy, as they sought synonymies for taxon and place that may not have been visible in this initial step.

	GBIF	VertNet	REMIB	UNIBIO	
Raw records	2,426,732	299,280	584,569	29,348	
Unique records	1,917,800	226,004	431,240	23,446	
Percent reduction	21.0	24.5	26.2	20.1	

Data reduction and cleaning

Initial data downloads totaled tens to hundreds of thousands of records from Mexico from each of the data portals, reaching millions from GBIF (although the GBIF numbers are largely from eBird and aVerAves, which come in greatest part from post-1980; Fig. 1, Table 1). We expected considerable redundancy between data sources, in the form of situations in which the same species was collected or recorded at the same place on the same day, so we embarked on a lengthy process of data reduction and cleaning. Without a doubt, some mistakes were made, and some information was lost, but this effort aimed to detect and highlight the major features of Mexican bird DAK, rather than all of the details. That is, we focused on sites that had the most information available, and explicitly excluded information for less-well-known sites.

A first step was to concatenate all four datasets into a single, larger dataset, and to reduce the set of fields to the essential three suites of fields mentioned above: species (we retained order, family, genus, and species, to permit identification of the most difficult names), date collected (year, month, and day), and place (latitude and longitude, when available, and state, municipality, and specific locality). We filtered these records to remove all those records from years after 1980. The initial total of 2,598,478 records distilled to 845,658 (a 67.5% reduction) that both (1) had dates and (2) were not from after 1980.

From this set, we extracted 10,762 unique combinations of order, family, genus, and species, which included diverse name combinations, and required considerable work to distill to a consistent suite of names corresponding to the birds of Mexico. We avoided the temptation to attempt to take the taxonomic treatment to a newer authority list (e.g., Gill & Donsker, 2014; Navarro-Sigüenza & Peterson, 2004; Peterson & Navarro-Sigüenza, 2006) because taxonomic splitting, which has dominated recent years of taxonomic work, would cause considerable confusion of names applied to older records. Hence, we reduced the initial, highly redundant set of names to 1,027 names that coincided with the taxonomy of the American Ornithologists’ Union (AOU, 1998), except that we merged Empidonax traillii and E. alnorum, in light of very frequent confusion in identification (Heller et al., 2016). This step was achieved based on long years of experience with Mexican bird taxonomy, plus occasional consultation of the literature (Monroe & Sibley, 1993; Peters, 1931–1987) and online (http://avibase.bsc-eoc.org/) resources.

A second step was to summarize date information as a concatenation of year, month, and day (e.g., 1964_7_16). Records with dates for which all three time elements were missing were removed from analysis, but records with partial dates (e.g., year and month, but not day) were treated as a unique time event. All data management was carried out in OpenRefine (http://openrefine.org), which permitted many important initial steps of combining similar names, and in Microsoft Access, which permitted development of customized queries for further refinement.

Data analysis

Perhaps the most difficult-to-manage of the fields was that of ‘place.’ Here, we used a three-step process that aimed to retain a maximum of locality information, yet avoid the massive and prohibitive task of full georeferencing of all records for which no geographic coordinates were available. Hence, (1) we used the 502,935 records that had latitude-longitude data in a first-pass analysis that aimed to identify single sites that were well inventoried, and to identify somewhat broader ‘hotspots’ of sampling (details provided below). Next (2), we used locality names associated with the records falling in the hotspots to probe the data lacking georeferences, and thereby rescued >54,000 records. Finally (3), we inspected the remaining data records—those lacking georeferences—to identify additional sites that merited analysis (five such additional, un-georeferenced sites indeed proved to be relatively well-inventoried, such that this step was important; see below). In this way, the number of localities that needed to be georeferenced was minimized, and yet we managed to include the great bulk of Mexican DAK in our analyses.

Step 1: Here, we aimed to develop first analyses of the 502,935 records from the original data set that carried latitude-longitude coordinates, both to identify individual pixels (0.05° resolution) that were well-inventoried, and to identify concentrations (“hotspots”) of sampling that may or may not prove to be well-inventoried. We first filtered this data set to retain only the records that were within 0.05° (5–6 km) of the administrative outline of Mexico—this step left 499,794 (99.4%) records for initial analysis. This initial data set was further reduced to 481,409 (95.7%) that came from 1980 or before.

We then created a shapefile with square elements of 0.05° for Mexico, and eliminated pixels that were >0.05° from the administrative boundaries of Mexico. We then used a spatial join to count numbers of records in each polygon; numbers of records per cell ranged from nil to as high as 15,464 (a pixel centered on Chilpancingo, Guerrero). We used optimized hot spot analysis (implemented in ArcGIS, version 10.2) to identify concentrations of sampling effort across the country—we used the Getis-Ord Gi* statistic (Hot Spot Analysis tool), and focused only on hotspots (0.05° grid squares) significant at the highest (99%) confidence level. We isolated these well-sampled suites of pixels as a separate shapefile, and then merged pixels in contiguous sets as preliminary hypotheses of broader hotspots of sampling, which we enriched with more records in step 2 (see below).

We also used the initial data set to develop an identification of single 0.05° pixels that were well-inventoried, as follows. The 481,409 pre-1980 records remaining after clipping to the country boundaries reduced to 277,249 (57.6%) unique combinations of species × date × pixel. We calculated Sobs as the number of species that have been recorded from each pixel, N as the total number of unique combinations of species × date from each pixel (and thus we set a base spatial resolution as that of the 0.05° grid, such that co-occurrences of species within these pixels are assumed to be sympatric), and a and b as the numbers of species recorded exactly once and exactly twice, respectively, from each pixel. We then used these data to calculate, for each pixel, the Chao2 estimator of expected species richness, with its associated adjustment for small sample sizes (Colwell, 1994–present), as (1) Sexp=Sobs+N−1Naa−12b+1.

Completeness was then calculated as C = Sobs / Sexp. To avoid including occasional localities with small sample sizes and artificially ‘complete’ inventories, we further removed all pixels for which N < 200. We summarized these calculations in a table, which we then imported into ArcGIS and joined to the grid shapefile for visualization.

Step 2: This step aimed to assess the fairly large portion of the data that lacked georeferences (195,371 records), and ‘rescue’ relevant data for further enrichment of the hotspot analysis. That is, in Step 1, we used georeferenced occurrence data to identify hotspots of sampling, but many more occurrences were documented in records lacking such data. Hence, here, first, we used the merged hotspot shapefile to associate the original occurrence data (i.e., records with georeferences) to hotspots, and identified key locality names among those occurrence data. (e.g., the Comitán hotspot included localities such as “La Trinitaria”, “El Triunfo,” “Lagunas de Montebello,” “Las Margaritas,” and “Santa Rosa”). Finally, we used those names to probe the non-georeferenced data records (checking, of course, that offset distances were not >5 km), and thereby rescued 54,369 records that were added to the hotspot-based analyses. Inventory statistics and completeness values were then recalculated for each hotspot just as they were calculated in Step 1 for each pixel.

Step 3: This final step involved review of the remaining non-georeferenced records, even after the rescue of Step 2. That is, Step 2 focused on non-georeferenced records that corresponded to already-identified pixels and hotspots, and that could be used to enrich the existing (georeferenced) data from those sites. This third step, however, involved a look at the remaining data to see if additional well-known sites could be identified.

We assembled the raw locality descriptors, and tallied numbers of records associated with each; we did some minor cleaning and synonymizing of minor variants on locality descriptors to maximize numbers of records for each locality. Finally, we developed completeness indices (as described above) for each of the non-georeferenced localities that had a raw sample size (i.e., unique combinations of locality descriptor, year/month/day, and species) of ≥200 records. Because the focus of these exercises is on detecting the few well-inventoried sites, rather than characterizing the overall sampling landscape, no biases are introduced by this rescue step. All localities meeting an arbitrary criterion of completeness, C ≥ 0.8, were then georeferenced and included among well-known sites.

Data availability

All data managed in this study are openly available, or will be shortly, as institutional permissions are finalized, from GBIF (http://www.gbif.org), VertNet (http://www.vertnet.org/), REMIB (http://www.conabio.gob.mx/remib/doctos/remib_esp.html), and UNIBIO (http://unibio.unam.mx/). A full synonymy of locality names in relation to the hotspots identified in this paper is available at http://hdl.handle.net/1808/20674. GIS shapefiles showing the two sets of hotspots identified in this paper are available at http://hdl.handle.net/1808/20673.

Results

Processing data available for birds of Mexico from raw DAK downloads into usable records of species at localities on particular days involved considerable reduction in numbers of records (Table 1). Simply progressing from raw to unique combinations of species × locality × geographic coordinates × year × month × day involved a reduction of 20.1–26.2% in numbers of records (note that this reduction step was done prior to removing records post-1980). Subsequent data cleaning and reduction steps (see Methods, above) reduced redundancy both among data sources (GBIF, VertNet, REMIB, UNIBIO) and among nearby localities, leaving a final number of 481,409 records for analysis.

Table 2 Summary of individual 0.05°grid squares that are well inventoried (C ≥ 0.8) across Mexico.

Names of grid squares refer to the shapefile dataset summarizing the geographic distribution of these sites.

State	Name	Latitude	Longitude	N	Sobs	a	b	Sexp	C	
Baja California	Isla Cedros	28.208	−115.244	213	45	12	10	51.0	0.88	
Isla San Benito	28.308	−115.594	398	60	19	11	74.2	0.81	
Baja California Sur	El Triunfo 2	23.808	−110.094	267	56	15	9	66.5	0.84	
La Paz 2	24.158	−110.244	1,167	137	35	34	154.0	0.89	
Laguna San Ignacio 2	27.308	−112.894	339	55	17	11	66.3	0.83	
San Jose de Cabo	23.058	−109.694	1,339	174	47	25	215.5	0.81	
Sierra de la Laguna	23.458	−109.794	314	56	9	5	62.0	0.90	
Sierra de la Laguna 4	23.558	−109.944	813	55	14	6	68.0	0.81	
Chiapas	El Fénix	16.658	−93.994	258	68	14	12	75.0	0.91	
Escuintla / Finca La Esperanza	15.358	−92.644	1,362	229	45	31	259.9	0.88	
Laguna Ocotal	16.808	−91.444	531	125	21	30	131.8	0.95	
Pueblo Nuevo	17.158	−92.894	480	133	39	28	158.5	0.84	
San Cristóbal de las Casas	16.658	−92.594	339	69	17	11	80.3	0.86	
San Cristóbal de las Casas 3	16.708	−92.694	342	65	18	11	77.7	0.84	
Chihuahua	Arroyo del Alamo	29.458	−106.794	214	76	21	17	87.6	0.87	
Arroyo el Mesteo	29.408	−106.894	208	76	22	21	86.4	0.88	
Colonia Pacheco	28.658	−106.194	283	63	17	9	76.6	0.82	
Colonia Pacheco 2	30.108	−108.294	945	103	21	9	124.0	0.83	
Rancho Pinos Altos	28.258	−108.294	259	49	8	5	53.6	0.91	
Rio El Gavilan	30.008	−108.394	325	83	19	15	93.7	0.89	
Ciudad de México	Xochimilco	19.258	−99.094	229	83	30	21	102.7	0.81	
Coahuila	Sabinas	27.858	−101.144	625	114	28	17	135.0	0.84	
Colima	Cihuatlan	19.208	−104.544	354	89	24	15	106.2	0.84	
Durango	El Salto	23.758	−105.544	205	55	12	16	58.9	0.93	
Neveros	23.758	−105.744	264	65	16	15	72.5	0.90	
Nievero	24.008	−104.744	215	52	12	12	57.1	0.91	
Santa Gertrudis	23.558	−104.394	254	86	27	17	105.4	0.82	
Santa Teresa	22.608	−104.844	264	68	16	10	78.9	0.86	
Tamazula	24.958	−106.944	225	99	37	29	121.1	0.82	
Guanajuato	Rancho Enmedio	21.108	−101.194	401	82	22	14	97.4	0.84	
Guerrero	Acahuizotla 2	17.358	−99.394	476	129	41	26	159.3	0.81	
Ajuchitlán	18.158	−100.494	219	40	9	4	47.2	0.85	
Chilpancingo	17.558	−99.494	6,530	270	50	22	323.3	0.84	
Cuapongo	17.508	−99.644	1,443	169	48	30	205.4	0.82	
Omiltemi 2	17.558	−99.644	2,419	206	47	30	240.9	0.86	
Teotepec	17.458	−100.194	254	47	10	6	53.4	0.88	
Jalisco	Atoyac	20.008	−103.594	316	107	31	24	125.5	0.85	
Atoyac 2	20.008	−103.544	223	72	20	15	83.8	0.86	
Lagos de Moreno 2	21.508	−101.694	202	70	24	15	87.2	0.80	
Tapalpa	19.958	−103.744	288	102	34	22	126.3	0.81	
Michoacán	Apatzingan	19.158	−102.444	244	68	18	12	79.7	0.85	
Pátzcuaro	19.458	−101.594	361	111	35	23	135.7	0.82	
Rancho El Bonete	18.958	−101.894	297	95	30	20	115.6	0.82	
Tzitzio	19.608	−100.944	285	78	19	13	90.2	0.87	
Tzitzio 2	19.658	−100.894	299	97	32	22	118.5	0.82	
Uruapan	19.408	−101.994	391	94	26	14	115.6	0.81	
Zacapu	19.808	−101.794	475	126	34	30	144.1	0.87	
Morelos	Cuernavaca	18.908	−99.244	541	159	50	32	196.1	0.81	
México	East of Zitácuaro	19.408	−100.194	250	77	22	14	92.3	0.83	
Puerto Lengua de Vaca	19.258	−99.894	224	63	14	14	69.0	0.91	
Temascaltepec	19.058	−100.044	632	128	29	21	146.4	0.87	
Nayarit	Islas Tres Marías	21.458	−106.444	394	64	15	15	70.5	0.91	
Islas Tres Marías 3	21.458	−106.394	475	68	20	12	82.6	0.82	
San Blas 3	21.558	−105.294	701	235	75	50	289.3	0.81	
Sauta	21.708	−105.144	370	107	34	20	133.6	0.80	
Tepic	21.258	−104.644	207	69	18	20	76.3	0.90	
Tepic 2	21.508	−104.894	1,126	178	43	22	217.2	0.82	
Oaxaca	Cerro San Felipe	17.158	−96.694	236	66	16	10	76.9	0.86	
Chivela	16.708	−94.994	251	73	20	16	84.1	0.87	
Palomares	17.108	−95.044	782	213	67	44	262.1	0.81	
Rancho San Carlos	17.208	−94.944	340	127	42	28	156.6	0.81	
Rio Molino	16.058	−96.444	374	84	17	16	92.0	0.91	
Totontepec	17.258	−96.044	533	107	15	30	110.4	0.97	
Quintana Roo	Felipe Carrillo Puerto	19.558	−88.044	212	86	26	31	96.1	0.89	
Felipe Carrillo Puerto 2	19.608	−88.044	342	131	45	37	157.0	0.83	
Isla Cozumel 4	20.508	−86.944	650	154	17	49	156.7	0.98	
Sinaloa	Babizos	25.758	−107.444	472	81	20	10	98.2	0.82	
Rancho Liebre	23.558	−105.844	601	126	26	21	140.7	0.90	
Sonora	Babizos 2	27.008	−108.394	539	106	29	18	127.3	0.83	
Chinobampo	26.958	−109.294	215	74	22	22	84.0	0.88	
Hacienda de San Rafael	27.108	−108.694	220	61	18	14	71.2	0.86	
Huasa	28.608	−109.794	221	66	21	12	82.1	0.80	
La Chumata	29.908	−110.594	241	68	15	17	73.8	0.92	
Oposura	29.808	−109.694	494	124	27	24	138.0	0.90	
Rancho Guirocoba 2	26.958	−108.694	879	168	39	27	194.4	0.86	
Tecoripa	28.608	−109.944	242	75	21	18	86.0	0.87	
Tamaulipas	Above Ciudad Victoria	23.708	−99.244	202	75	2	49	75.0	1.00	
Ciudad Victoria	23.708	−99.144	370	177	52	33	215.9	0.82	
Gomez Farías 3	23.058	−99.094	354	123	44	32	151.6	0.81	
Matamoros	25.858	−97.494	628	207	60	42	248.1	0.83	
Tampico	22.258	−97.844	1,038	211	47	41	236.7	0.89	
Veracruz	Balzapote 4	18.608	−95.044	3,491	286	56	43	321.0	0.89	
Cerro Guzman	19.208	−96.394	542	141	43	29	171.0	0.82	
Coatepec	19.458	−96.944	377	122	16	49	124.4	0.98	
Coyame	18.408	−94.994	338	129	30	41	139.3	0.93	
Huatusco	19.158	−96.944	282	95	26	30	105.4	0.90	
Lago de Catemaco	18.408	−95.144	205	79	28	21	96.1	0.82	
Misantla	19.908	−96.844	307	108	3	62	108.0	1.00	
Perote	19.558	−97.244	263	88	14	44	90.0	0.98	
Presidio 2	18.658	−96.744	1,000	219	54	36	257.6	0.85	
Rancho Caracol 3	18.458	−96.644	358	127	41	26	157.3	0.81	
Xalapa	19.558	−96.944	887	192	34	54	202.2	0.95	
Yucatán	Chichén-Itzá	20.658	−88.594	1,727	193	26	23	206.5	0.93	
Mérida	20.958	−89.644	242	118	38	25	144.9	0.81	
Xocompich	20.758	−88.544	270	115	44	40	138.0	0.83	

Figure 2 Distribution of single 0.05° grid squares for which >200 records were available and completeness (C) was 0.8 ≤ C < 0.9 (pink) or C ≥ 0.9 (brown).

Sites detected in Step 3 (i.e., single sites that are relatively complete, but that were not georeferenced prior to this study) are shown in blue (all had 0.8 ≤ C < 0.9).

A first analysis focused on 277,249 unique combinations of species × date collected × 0.05° pixel. These records fell in pixels in numbers ranging from nil to 15,464 records (Chilpancingo, Guerrero). Processing records in each pixel into estimates of expected numbers of species and estimated inventory completeness (Table 2; Fig. 2), we found that 24 pixels were complete at the level of C ≥ 0.9, and a further 71 pixels were complete at the level of C ≥ 0.8. The well-inventoried sites were well-distributed across the country, from Baja California Sur and Sonora to Quintana Roo and Chiapas (Fig. 2). A further five localities were rescued from among the pool of data lacking geographic coordinates, but that were inventoried completely to the level of C ≥ 0.8 (Table 3).

Seeking ‘hotspots’ of sampling (i.e., sets of contiguous 0.05° pixels), we identified an initial large number of such hotspots, again well-distributed across the country. Our inspection of the data lacking geographic coordinates ‘rescued’ 54,369 records, augmenting the initial data set considerably. Of the initial sampling hotspots, only three made the C ≥ 0.9 completeness criterion (Xalapa, Veracruz; El Triunfo, Chiapas; Isla Cozumel, Quintana Roo); another 17 fell at the lower C ≥ 0.8 criterion (Table 4). These hotspots covered areas ranging from 60.5–4325.8 km2, considerably larger than the ∼30 km2 of the 0.05° pixels. The hotspots, once again, ranged across the country, from southern Sonora to Chiapas and Quintana Roo (Fig. 3).

Discussion

This contribution is an exploration of the utility of the existing Digital Accessible Knowledge (Sousa-Baena, Garcia & Peterson, 2013) for Mexican birds in identifying well-inventoried sites across the country. We chose our 1980 cut-off to coincide roughly with the transition between initial (subtle) climate changes and the present phase of rapid, large-scale climate change worldwide (IPCC, 2013). In this way, the sites that we have identified provide baseline points of reference for species composition, for comparison with species composition later, decades into the processes of global climate change (Tingley et al., 2009).

One could map the species diversity that has been documented or that we have estimated for 0.05° grid squares and/or hotspots across the country, to obtain a picture of species diversity countrywide. We have avoided this temptation, however, in view of the highly non-random and scattered distribution of the well-inventoried sites across the country. The sites do not cover all regions of the country at all consistently, so any the results would be incomplete and potentially misleading. That step was, quite simply, not among the objectives of this study.

Rather, our aim was to compile a catalog of sites across Mexico that have been inventoried historically in detail, to the point that the inventory is more or less complete. This catalog, in and of itself, is not of great interest scientifically; however, to the extent that new inventories can be developed for comparison with the old ones, the interest in the comparisons grows considerably. Echoing our earlier contribution (Peterson et al., 2015), we are fascinated by the long-term processes of population biology and biogeography that are leading to turnover of species at sites.

Table 3 Summary of additional sites that were ‘rescued’ from among digital data lacking geographic coordinates, but that were detected based on unique locality descriptors.

State	Locality name	Latitude	Longitude	N	Sobs	a	b	Sexp	C	
Nuevo León	Monterrey	25.687	−100.316	418	145	52	38	178.9	0.81	
Tabasco	1 mi E Teapa	17.563	−92.948	361	158	57	41	195.9	0.81	
San Luis Potosí	[Ciudad] Valles	21.997	−99.011	227	93	34	26	113.7	0.82	
Chiapas	26 km N by road Ocozocuautla	16.995	−93.379	400	145	55	40	181.1	0.80	
Oaxaca	1 mi SW Valle Nacional	17.757	−96.320	485	169	54	46	199.4	0.85	

Table 4 Hotspots of sampling that are relatively completely inventoried (i.e., C ≥ 0.8). Hotspot names correspond to the shapefile dataset summarizing the geographic distribution of these sites.

State	Hotspot	N	Sobs	a	b	Sexp	C	
Chiapas	Arriaga Tonalá	2,162	337	88	53	407.86	0.83	
Comitán	3,821	422	79	62	470.9	0.89	
El Triunfo	4,884	396	41	34	419.4	0.95	
Tuxtla Gutiérrez	4,058	463	93	54	540.8	0.86	
Ciudad de México	Valle de México	6,818	349	71	28	434.7	0.80	
Guerrero	Chilpancingo / Omiltemi	15,230	380	51	24	431.0	0.88	
Michoacán	Patzcuaro / Morelia / Lagos de Michoacán	2,144	269	60	26	334.5	0.80	
Tancítaro / Uruapan	1,396	253	50	28	295.2	0.86	
Nayarit	Tepic / San Blas	5,079	405	66	34	466.3	0.87	
Oaxaca	Matías Romero to north	3,994	404	84	42	485.0	0.83	
Miahuatlán / Mixtepec	1,054	205	54	34	245.9	0.83	
Tapanatepec / Zanatepec / Cerro El Baúl	4,752	355	63	28	422.3	0.84	
Quintana Roo	Isla Cozumel	2,159	216	39	46	231.8	0.93	
Sinaloa	Durango Hwy / Espinazo del Diablo	3,857	293	57	24	356.8	0.82	
Rosario	1,912	247	55	35	288.2	0.86	
Sonora	Álamos	3,403	273	42	23	308.9	0.88	
Veracruz	Los Tuxtlas	10,499	457	78	41	528.5	0.87	
Orizaba / Córdoba	2,741	402	94	52	484.4	0.83	
Xalapa	2,135	303	54	49	331.6	0.91	
Yucatán	Chichén-Itzá	2,612	220	37	20	251.7	0.87	

Figure 3 Summary of “hotspots” of sampling of Mexican avifaunas based on the Getis-Ord Gi* statistic.

Gray areas have 0.8 ≤ C < 0.9; red areas have C > 0.9.

We have improved on our earlier contribution (Peterson et al., 2015), however, which had to aggregate occurrence data to coarse resolutions (1°, or ∼110 km) until inventories were sufficiently complete. Here, in contrast, we have sought single sites (0.05° grid squares) that are completely inventoried—this difference has the major advantage of not including as much beta diversity in single-site inventories as did our previous study. Our consideration of hotspots, to some degree, began to coarsen the spatial extent of the sites once again, but offered a somewhat more extensive list of sites that have seen thorough inventories, yet still across extents much smaller than in our previous work.

One criticism that can be leveled at this work is that some important data may have been left out of the analysis. That is, the entire concept of Digital Accessible Knowledge is that the data are (1) in digital format, (2) accessible readily via the Internet, and (3) integrated with the remainder of DAK via common portals (i.e., making the transition from individual data points to integrated “knowledge”), as was emphasized in the original publication presenting the idea of DAK (Sousa-Baena, Garcia & Peterson, 2013). (We note that a subsequent publication (Meyer et al., 2015) used “digital accessible information,” we believe unfortunately, as they provided no justification for or even notice of the change of terminology.) In the case of Mexican birds, for example, the Natural History Museum (UK) has very few data in digital format, and none has been made accessible, such that important collections from Mexico, like those assembled by Godman & Salvin (1879–1915), have not been analyzed in this contribution. That is the blessing and the curse of DAK: data that are digital and shared on global portals are used broadly, whereas data that do not meet the DAK criteria are frequently not used at all.

The purpose of this paper is to enable a broad suite of repeat avifaunal survey efforts across Mexico. In effect, with the maps and tables of this paper, we challenge the ornithological community interested in Mexican birds to focus attention on these sites (we provide a complete compendium of the well-inventoried sites and hotspots documented in this paper, as well as associated shapefiles, in a dataset made available permanently at http://hdl.handle.net/1808/20673). Not only does work at these sites provide information about the current community composition there, but also about the change in those communities through time.

We suggest and advocate that resurvey efforts take the form of two inventories at or near the site: one in as exactly the original site as is possible to determine, and the other in the closest and most comparable site that still retains the vegetation type that was represented at the site at the time of the original inventory efforts; the former reflects effects of local-scale processes (e.g., habitat destruction, aridification), whereas the latter reflects more global processes (e.g., climate change), and comparisons of the two resurvey inventories will yield a rich understanding of the relative magnitude of effects of the local and global processes in changing avifaunas across the country. Once several such sites have been re-surveyed, a rich picture of the dynamics of Mexican bird distributions will emerge, in much greater detail than the picture presently available (Peterson et al., 2015).

Figure 4 Photos of landscapes of two of the hotspots identified in this study from the Nelson-Goldman expeditions across Mexico in the late nineteenth and early twentieth centuries: above Xalapa, Veracruz (SIA2014-03203), Mt. Tancítaro, Michoacán (SIA2016-03203).

Photos reproduced with permission of the Smithsonian Institution, Washington, D.C.

We are working to assemble what historical photographs exist for each of these sites, in association with the original inventory efforts, to further enrich comparisons of ‘before’ and ‘after.’ For instance, the Nelson and Goldman expeditions across Mexico in the late nineteenth and early twentieth centuries produced large quantities of images that are well-documented in two summary volumes (Goldman, 1951; Nelson, 1921)—two examples are shown in Fig. 4. A major complement to the repeat inventories that are facilitated by the analyses in this study would be repeat photography to allow a clear view of what sorts of landscape change have occurred at key sites (see, e.g., Spond, Grissino-Mayer & Harley, 2014).

Conclusions

Valuable insights can be gained from longitudal comparisons to detect and characterize patterns of change in biodiversity, yet such changes have been opaque to study for lack of paired temporal samples at sites or in areas. This study explores a novel approach to enabling such studies: we mine the existing DAK for Mexican birds to detect and document well-inventoried sites, and provide a catalog of those well-known sites for others to use. Developing repeated inventories at a series of sites across Mexico would yield a detailed, controlled set of comparisons that would allow a view of avifaunal dynamics across the country. Adding the dimension of repeated landscape photography to the repeated inventories would greatly facilitate pairing of sites for future inventories at sites for which historical information exists.

We thank the Instituto de Biología and the Comisión Nacional para el Uso y Conocimiento de la Biodiversidad for generously supplying access to data.

Appendix

Institutions contributing data to the analyses reported in this paper: Academy of Natural Sciences of Philadelphia; American Museum of Natural History; Angelo State University; Australian National Wildlife Collection; Bell Museum of Natural History; Biologiezentrum Linz; Bird Studies Canada; Borror Laboratory of Bioacoustics; Burke Museum, University of Washington; California Academy of Sciences; California State University, Chico; Canadian Museum of Nature; Carnegie Museum of Natural History; Chicago Academy of Sciences; CIIDIR-Oaxaca; Comisión Nacional para el Uso y Conocimiento de la Biodiversidad; Cornell Laboratory of Ornithology; Delaware Museum of Natural History; Denver Museum of Nature & Science; ECOSUR Chetumal; ECOSUR San Cristóbal; Emporia State University; Estación Biológica Doñana; FES Zaragoza UNAM; Field Museum of Natural History; Florida Museum of Natural History; Fort Hays State University; HawkCount; Humboldt State University; Illinois State University; iNaturalist; INIFAP; Instituto de Biología, Universidad Nacional Autónoma de México; Instituto de Ecología, A.C., Xalapa; Instituto de Historia Natural de Chiapas; Instituto Humboldt, Colombia; Louisiana State University; Lund Museum; Michigan State University; Moore Laboratory of Zoology; Musée de la Vallée, Barcelonette, France; Musée George Sand et de la Vallée Noire; Museo de las Aves de México; Museo de Zoología ”Alfonso L. Herrera,” Facultad de Ciencias, Universidad Nacional Autónoma de México; Museo Nacional de Ciencias Naturales; Museu Paraense Emilio Goeldi; Museum für Naturkunde, Berlin; Museum Heineanum Halberstadt; Museum National d’Histoire Naturelle; Museum of Comparative Zoology, Harvard University; Museum of Evolution, Uppsala; Museum of Nature and Human Activities, Hyogo, Japan; Museum of Southwestern Biology; Museum of Vertebrate Zoology; Museum Victoria, Australia; Natural History Museum (Bird Group, Tring); Natural History Museum of Los Angeles County; Naturalis, Amsterdam; Neotropical Ornithological Foundation; New York State Museum; North Carolina Museum of Natural Sciences; Ocean Biodiversity Information System; Ohio State University; Orcutt Trust Collection; Perot Museum of Nature and Science; Polish Academy of Sciences; Provincial Museum of Alberta; Queensland Museum; Royal Belgian Institute of Natural Sciences; Royal Ontario Museum; San Diego Natural History Museum; Santa Barbara Museum of Natural History; Senckenberg Museum; Slater Museum of Natural History; South Australian Museum; Staatliche Naturhistorische Sammlungen Dresden, Museum für Tierkunde; Staatliches Museum für Naturkunde, Stuttgart; Tall Timbers Research Station and Land Conservancy; Texas A&M University; Tulane University; U.S. National Museum of Natural History; Uberseemuseum, Bremen; Universidad Autónoma de Baja California; Universidad Autónoma de Baja California Sur; Universidad Autónoma de Campeche; Universidad Autónoma de Nuevo León; Universidad Autónoma de San Luis Potosí; Universidad Autónoma de Tamaulipas; Universidad Autónoma del Estado de Hidalgo; Universidad Autónoma del Estado de México; Universidad Autónoma del Estado de Morelos; Universidad de Ciencias y Artes de Chiapas; Universidad de Guanajuato; Universidad de Navarra; Universidad Juárez Autónoma de Tabasco; Universidad Juárez del Estado de Durango; Universidad Michoacana de San Nicolás de Hidalgo; University Museum of Zoology, Cambridge; University of AL; University of Alaska Museum; University of Alberta; University of Arizona; University of British Columbia; University of California, Davis; University of California, Los Angeles; University of Colorado; University of Iowa; University of Kansas; University of Michigan Museum of Zoology; University of Nebraska State Museum; University of Oklahoma; University of Oslo; University of Texas El Paso; University of Wyoming; Utah Museum of Natural History; Washington State University; Western Foundation of Vertebrate Zoology; Western New Mexico University; Yale Peabody Museum; Yamashina Institute of Ornithology; Zoologische Staatssammlung München; and Zoologischen Sammlung der Universitat Rostock.

Additional Information and Declarations

Competing Interests

Author Contributions

Data Availability

The authors declare there are no competing interests.

A. Townsend Peterson conceived and designed the experiments, performed the experiments, analyzed the data, wrote the paper, prepared figures and/or tables, reviewed drafts of the paper.

Adolfo G. Navarro-Sigüenza conceived and designed the experiments, performed the experiments, prepared figures and/or tables, reviewed drafts of the paper.

Enrique Martínez-Meyer conceived and designed the experiments, performed the experiments, reviewed drafts of the paper.

The following information was supplied regarding data availability:

KU ScholarWorks (http://hdl.handle.net/1808/20674) provides a full synonymy of locality names in relation to the hotspots identified in this paper.

GIS shapefiles showing the two sets of hotspots identified in this paper can be found at http://hdl.handle.net/1808/20673.

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
