# Peer review of "Digital Accessible Knowledge and well-inventoried sites for birds in Mexico: baseline sites for measuring faunistic change"

_PeerJ, doi:10.7717/peerj.2362_

## Round 0.1 · original submission · Minor Revisions

The three referees make a number of complimentary remarks on the paper and suggestions to improve its presentation.

Reviewer 1 ·

Basic reporting

1. line 125: reason (line 246) to use choose cut off 1980 should come first

2. Methodology: it might confuse the audience and they will have to read more than once. Data reduction and cleaning, and analysis can be represented well in a schematic way and can be explained in the text precisely.

3. line 209: There is no justification on the selection of complete index =>0.8.

Experimental design

Methodology is profound, all the possible efforts are made to avoid bias.

Validity of the findings

no comments

Additional comments

Data mining is a very important step prior to analysis. and equally it is important to address related sampling bias. To this note, I think that the paper has not only achieved the objective but also clearly addressed the bias. Moreover, the steps to deal with such data are very clear and concise.

Reviewer 2 ·

Basic reporting

Language used throughout the manuscript is ambiguous in some cases. In addition, personal pronouns have been used throughout the manuscript.
According to academic/scientific standards [e.g. OWLL (Massey Univiersity), http://owll.massey.ac.nz/academic-writing/1st-vs-3rd-person.php ] “Traditional academic writing discourages the use of first or second person (‘I’, ‘we’, ‘you’, etc.). This is because it does not sound objective. Instead, it sounds as though you have only a very limited, personal view of the issue you are discussing, rather than a view of the broader picture”.
The introduction and background are appropriate to show the relevance of conducting research on the topic. However, edition of the text would be needed, that would help to enhance the cohesion of the manuscript.
Authors fail following some of the PeerJ standards for publication, comments are given on the manuscript.

Experimental design

The manuscript is original and meet the scope of the journal. In addition The research question is clearly defined and the manuscripts highlights a knowledge gap clearly.

Methods are described with sufficient detail and information that would allow not only replication of the analysis but also would stimulate further research on the subject. In addition, robust investigation has been conducted that the manuscript meets high technical and ethical PeerJ standards.

Validity of the findings

The manuscript contributes to assess the utility of existing digital accessible knowledge for Mexican birds in well-inventoried sites across the country, and provides and update of previous research conducted by the authors (Peterson et al. 2015). The novelty of the manuscript stems from the fact that the results presented in the manuscript depict well-inventoried areas across the country and these results would enhance the development of additional research on the topic.

The information presented in the analysis is robust and statically sound. It represents the wide picture of birds inventoried sites across Mexico.
The conclusion is well stated and linked to the original research question. However, is not limited to support the results presented in the study, speculation has been stated in the conclusion and not referred as such.

Annotated reviews are not available for download in order to protect the identity of reviewers who chose to remain anonymous.

·

Basic reporting

The article meets the appropriate standards. The text is clear and leaves little room for misinterpretation except for a couple of minor points that have been commented out in the text. Background and introductory materials, rationale, scope and objectives are clearly stated. Figures are absolutely necessary and tables do provide verification data. Additional results are provided as separate files. Although the actual raw data are not apparently provided as a file at the time of review, they can be obtained from mostly public sources.

Experimental design

This research is original and the methods are sound and fully conform to the state-of-the-art. There are some steps that may require some further clarification and these have been spelled out as comments in the attached text.

Validity of the findings

The findings, as based in the raw data used, seem correct and valid. Arguably though, some of these findings might be open to debate or could be challenged by additional data; see comments in text for details. In any case, to my knowledge this research does not replicate any previously published research and is offering new results that are of scientific worth.

Additional comments

See specific comments in the revised text. See also some typo corrections.

In addition, two general comments, which may be further expanded in the text comments:

- Baselines are defined as existing in DAK sources, and these are defined in a highly restrictive fashion. However the aim of the study was stated as listing all sites for which enough pre-1980 data existed, and the summary as it is conveys this idea. So other sources of (possibly un- or under-digitized) data should also be considered, e.g. reports or management plans for reserve biospheres, and not only DAK, if the aim is to list all such sites--otherwise the study should clearly define itself as a work on, and limited to, sensu-stricto DAK and not intend to be broader than this.

- The basics of the study rely on location-time-taxon strings, of which location is collapsed into spatial cells--thereby leaving differences in time as occurrences for a taxon. A date analysis should show whether there were preferred but possibly invalid detailed dates e.g. first of the month. The Methods section should consider whether this was done or not. Dates missing the day were deemed valid (as a single date string) but should be distinguishable from the "first-of-the-month" as representative from the entire month. If two of those occur at a given place and for a given taxon they should preferably be treated as one date only.

---

## Round 0.2 · accepted · Accept

Thank you for clearly addressing the referees suggesitons.